# C3PO: Critical-Layer, Core-Expert, Collaborative Pathway Optimization for Test-Time Expert Re-Mixing

**Zhongyang Li**
Department of Computer Science
Johns Hopkins University
zli300@jh.edu

**Ziyue Li**
Department of Computer Science
University of Maryland, College Park
litzy619@umd.edu

**Tianyi Zhou**
tianyi.david.zhou@gmail.com

## Abstract

Mixture-of-Experts (MoE) Large Language Models (LLMs) suffer from severely sub-optimal expert pathways—our study reveals that naive expert selection learned from pretraining leaves a surprising 10-20% accuracy gap for improvement. Motivated by this observation, we develop a novel class of test-time optimization methods to re-weight or "re-mixing" the experts in different layers jointly for each test sample. Since the test sample's ground truth is unknown, we propose to optimize a surrogate objective defined by the sample's "successful neighbors" from a reference set of samples. We introduce three surrogates and algorithms based on mode-finding, kernel regression, and the average loss of similar reference samples/tasks. To reduce the cost of optimizing whole pathways, we apply our algorithms merely to the core experts' mixing weights in critical layers, which enjoy similar performance but save significant computation. This leads to "**C**ritical-Layer, **C**ore-Expert, **C**ollaborative **P**athway **O**ptimization (C3PO)". We apply C3PO to two recent MoE LLMs and examine it on six widely-used benchmarks. It consistently improves the base model by 7-15% in accuracy and outperforms widely used test-time learning baselines, e.g., in-context learning and prompt/prefix tuning, by a large margin. Moreover, C3PO enables MoE LLMs with 1-3B active parameters to outperform LLMs of 7-9B parameters, hence improving MoE's advantages on efficiency. Our thorough ablation study further sheds novel insights on achieving test-time improvement on MoE. Our code can be accessed here.

## 1 Introduction

Mixture-of-Experts (MoE) has achieved remarkable success when being extended to recent large language models (LLMs). By only selecting one (or a few) out of $N$ experts in each layer, MoE LLMs can reduce their activated parameters to $1/N$ during inference while keeping their model capacity the same as models of the same size, thereby providing a more efficient scaling law in practice (Lepikhin et al., 2020; Fedus et al., 2022). In MoE LLMs, the sequence of expert choices or weights across multiple layers, i.e., the so-called "pathway", differs across samples and is generated by routers or gates trained together with other model parameters in an end-to-end manner. The pathway determines the experts to apply in each layer and thus greatly impacts the final performance. However, we find that the pathways generated by routers in existing MoE LLMs are prone to severe sub-optimality on

various samples/tasks, leading to a large gap (10-20%) between base model and the oracle with optimal pathways, as shown in Table 1. This implies a large room for improvement that existing approaches have not explored.

Although test-time optimization and adaptation on large language models (LLMs), e.g., in-context learning (ICL) (Brown et al., 2020), prompt/prefix tuning (Lester et al., 2021; Li & Liang, 2021), etc., have been widely studied, showing great potential of enhancing downstream task performance without finetuning any pre-trained parameters, it is still an open problem on MoE/Pathway LLMs what test-time optimization can effectively enhance the adaptation performance. Motivated by the observed sub-optimality of pathways and their routing weights, we propose to optimize the pathways for each test sample/task. Compared to prompt/prefix tuning, pathway optimization only needs to optimize much fewer variables (e.g., tens to hundreds of expert routing weights) than prompt/prefix, in which every token is composed of thousands of dimensions. Compared to ICL, which requires a large memory of exemplars yet still suffers from high variance of exemplar selection, pathways are much more compact representations describing how an MoE LLM addresses each task using different experts at different stages. Moreover, due to the relatively low dimensions of pathways, it is possible to avoid gradient-based backpropagation and instead rely on much more efficient gradient-free search.

To this end, we explore three pathway optimization approaches developed for test-time adaptation, all leveraging reference pathways for a few successful samples/tasks close to the test sample/task collaboratively. Ideally, a test sample's optimal pathway minimizes its loss on the model output (oracle). However, since the test sample's ground truth is unknown, we resort to its nearest neighbors in a reference set of samples associated with pathways leading to correct responses. Specifically, we optimize a surrogate objective as (1) mode finding in the space of pathways; (2) kernel regression of pathway routing weights in the neighborhood; and (3) weighted sum of losses on nearest neighbors. While optimizing the first two objectives does not require backpropagation, gradient-based optimization is needed for the third. In our experiments, (1) achieves comparable performance to more expensive ICL and prompt/prefix tuning, while (2), especially (3), significantly outperforms them, demonstrating the advantages of **C**ollaborative **P**athway **O**ptimization (CPO) on both efficiency and performance.

Since a pathway still involves tens to hundreds of routing weights or expert choices to optimize, can we further reduce the optimization cost? To answer this question, we investigate the importance and contribution of layers and experts in CPO. Our analysis reveals that at most 5 layers suffice to achieve the best performance across all the evaluated downstream tasks, where optimizing the pathways in the last few layers usually performs the best among other combinations of layers, as shown in Figure 3. In addition, as recent sparse MoE LLMs have 64 experts per layer but only select the top-8 for each input, we investigate whether optimizing a small portion of experts' routing weights can cover the top-8 and retain the performance of all-expert pathway optimization. As revealed by Figures 4, 5 and 7, only optimizing the top 8-20 experts can preserve the top-8 and the performance of all-expert optimization.

Motivated by these empirical analyses, we propose "**C**ritical-Layer, **C**ore-Expert, **C**ollaborative **P**athway **O**ptimization (C3PO)" that focuses on optimizing pathways on critical layers for core experts. We apply C3PO to two SOTA MoE LLMs, i.e., OLMoE and DeepSeekMoE, and consistently achieve improvement of 7-15% over the base models in accuracy across six benchmarks. Moreover, C3PO enables the MoE LLMs with 1-3B active parameters to outperform LLMs with 7-9B parameters. Furthermore, we conduct a comprehensive ablation study of different choices in C3PO, such as optimized tokens, steps, neighbors, kernel, etc. C3PO shows great potential to thoroughly exploit the advantages of MoE/pathway LLMs in model capacity and inference efficiency.

## 2 Related Work

**MoE LLMs** MoE architectures have been widely adopted in LLMs to improve efficiency and specialization (Shazeer et al., 2017). Recent works such as OLMoE (Muennighoff et al.,

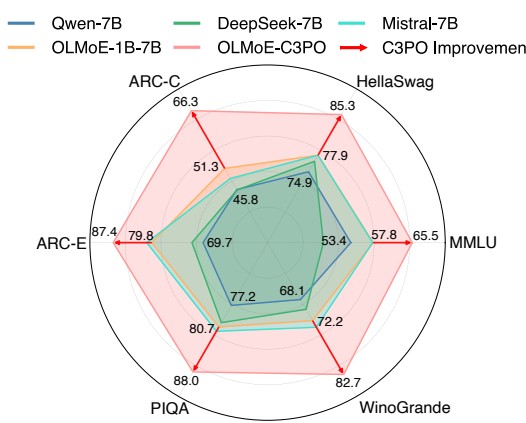

Figure 1: Comparison of OLMoE-1B-7B (1B activated parameters) with C3PO against multiple 7B dense models across six benchmarks. C3PO improves OLMoE-1B-7B's accuracy by 7-15%, outperforming 7B models over all benchmarks, validating the efficiency of MoE architecture and C3PO's optimization effectiveness.

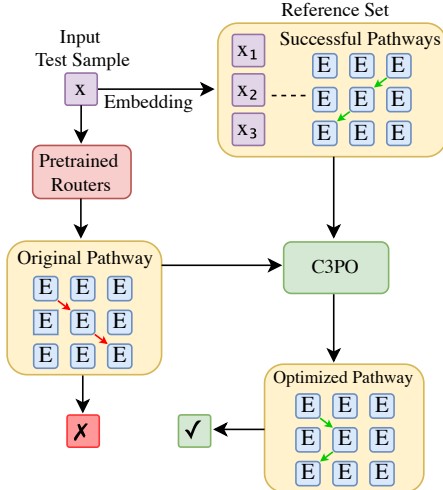

Figure 2: Pathway optimization in C3PO. For a test sample, C3PO retrieves successful pathways (green arrows) from similar samples in the reference set and adjusts the initial pathway (red arrow) based on them to achieve better prediction.

2024) and DeepSeekMoE (Dai et al., 2024) demonstrate the effectiveness of sparse MoE layers in reducing active parameters while maintaining model capacity. These models leverage token-choice routing to activate subsets of experts dynamically, enabling fine-grained specialization. The performance of MoE models heavily depends on expert selection mechanisms. Traditional routing strategies are trained end-to-end with the model (Fedus et al., 2022; Jiang et al., 2024), but our study reveals significant sub-optimality in these pathways.

**Efficient Adaptation of LLMs** Recent work has explored efficient adaptation of LLMs to downstream tasks with minimal computational overhead, aligning closely with our goal of efficient inference-time optimization. Among these approaches (Li et al., 2025a; Li & Zhou, 2024; Li et al., 2025c), In-Context Learning (Brown et al., 2020) appends task demonstrations to the input prompt to steer model behavior through attention mechanisms, avoiding weight updates but significantly increasing sequence length and memory requirements. Alternative methods like Prefix Tuning (Li & Liang, 2021) prepend trainable vectors to transformer layers to guide model outputs, while Prompt Tuning (Lester et al., 2021) learns continuous or discrete prompt tokens through gradient updates to embedding parameters. While these methods (Li et al., 2025b; Li & Zhou, 2025) share our objective of avoiding full parameter retraining, C3PO introduces two fundamental innovations. First, where existing techniques either modify model weights or substantially expand input length, our method preserves all original model parameters entirely while maintaining the standard input token budget. Second, rather than relying on static task-specific adaptations encoded through prompts or tuned parameters, we dynamically optimize routing weights for each test sample based on similarity to successful reference examples.

## 3 Methodology

MoE LLMs use routers to dynamically select and weight experts across layers, forming a specific pathway. However, these end-to-end trained routers often produce suboptimal pathways for challenging or out-of-distribution samples, which can significantly degrade the performance of MoE on diverse downstream tasks. The importance of expert pathways

has been broadly demonstrated on six benchmarks in our experiments: There exists a substantial performance gap between the base model (using the default expert pathways) and the oracle (using the optimal expert pathways) as shown in Table 1, revealing the potential benefits of optimizing expert pathways during inference.

To address this limitation, Critical-Layer, Core-Expert, Collaborative Pathway Optimization (C3PO) introduces a dynamic test-time re-mixing mechanism that adapts the pathway matrices for each test sample based on similar samples in a **reference set**—a collection of samples on which the MoE LLM's outputs are correct or preferred. Specifically, given a reference set of $m$ samples $\{(x_i, y_i)\}_{i=1}^{m}$ and their corresponding expert pathway matrices $\{\omega_i\}_{i=1}^{m}$ (where each $\omega_i \in \mathbb{R}^{L \times E}$, with $L$ denoting the number of layers and $E$ the number of experts) on which the model makes correct predictions (i.e., $f(x_i, \omega_i) = y_i$), for a new test sample $x$, the goal of C3PO is to find an improved expert pathway matrix $\omega$ for $x$ that leads to more accurate and higher-quality output $f(x, \omega)$.

## 3.1 Gradient Descent

We iteratively update $\omega$ using gradient descent:

$$\omega \leftarrow \omega - \lambda \nabla_{\omega} L(\omega), \tag{1}$$

where $\lambda$ is the learning rate and $L(\omega)$ is the objective function. Two variants are considered:

**Oracle (Upper Bound)** Assuming we know the ground truth label $y$ for $x$, we set

$$L(\omega) = \ell\big(f(x, \omega), y\big), \tag{2}$$

where $\ell(\cdot, \cdot)$ is the loss function (e.g., cross-entropy or L2 loss) measuring the discrepancy between model output $f(x, \omega)$ and ground truth $y$. Although impractical to have the ground truth in real scenarios, this method provides a performance ceiling to reveal the degradation caused by sub-optimal expert pathways and evaluate the effectiveness of other methods.

**Neighborhood Gradient Descent (NGD)** Without the truth label $y$ for $x$, we approximate the gradient of $\omega$ by using the loss functions of the nearest neighbors of $x$ in the reference set :

$$L(\omega) = \frac{\sum_{i \in \mathcal{N}(x)} K(x_i, x) \, \ell\big(f(x_i, \omega), y_i\big)}{\sum_{i \in \mathcal{N}(x)} K(x_i, x)}, \tag{3}$$

where $K(\cdot, \cdot)$ is the kernel function, e.g., Gaussian kernel, Matern kernel, etc. By leveraging loss information from the neighborhood of $x$, NGD establishes a test-time adaptation mechanism without accessing truth label $y$. This approach effectively aligns $\omega$ with the successful expert pathways in the reference set.

## 3.2 Kernel Regression

Kernel regression estimates the optimal expert pathways by computing a weighted average of the neighbors' expert pathway matrices:

$$\hat{\omega} \triangleq \frac{\sum_{i \in \mathcal{N}(x)} K(x_i, x) \, \omega_i}{\sum_{i \in \mathcal{N}(x)} K(x_i, x)}. \tag{4}$$

Although setting $\omega \leftarrow \hat{\omega}$ already improves performance in the experiments, we further refine the result by interpolating between the initial $\omega$ and $\hat{\omega}$:

$$\omega \leftarrow \alpha \, \omega + (1 - \alpha) \, \hat{\omega}, \tag{5}$$

with the optimal $\alpha$ chosen as

$$\alpha^* = \arg\min_{\alpha} L\big(\alpha \, \omega + (1 - \alpha) \, \hat{\omega}\big). \tag{6}$$

This refinement step balances the kernel regression estimate with the original expert pathway matrices.

### 3.3 Mode Finding (Meanshift)

Mode finding shifts $\omega$ toward the densest region of the mixing weight space to capture the most consistent routing patterns among neighbors. The update is performed as:

$$\omega \leftarrow \alpha\,\omega + (1-\alpha)\,\bar{\omega}, \tag{7}$$

where the local average $\bar{\omega}$ is computed in the $\omega$-space:

$$\bar{\omega} \triangleq \frac{\sum_{i \in \mathcal{N}(\omega)} K(\omega_i, \omega)\,\omega_i}{\sum_{i \in \mathcal{N}(\omega)} K(\omega_i, \omega)}. \tag{8}$$

Here, $\mathcal{N}(\omega)$ denotes the neighborhood defined in the expert pathway matrices space.

### 3.4 Neighborhood and Embedding Space

**Neighborhood** The neighborhood $\mathcal{N}(x)$ can be defined via $k$NN or $\epsilon$-ball:

$$\mathcal{N}(x) \triangleq \arg \min_{A \subseteq 2^m, |A| \le k} \sum_{i \in A} d(x_i, x), \tag{9}$$

$$\mathcal{N}(x) \triangleq \{i \in [m] : d(x_i, x) \le \epsilon\}, \tag{10}$$

where $d(\cdot, \cdot)$ is is defined as one minus the cosine similarity between the embedding vectors.

**Embedding Space** Instead of applying $K(\cdot, \cdot)$ and $d(\cdot, \cdot)$ directly on the raw inputs $x_i$ and $x$, we can replace $x$ and $x_i$ with their embedding $E(x)$ and $E(x_i)$, where $E(\cdot)$ is a pre-trained embedding model applied to the task description of each sample.

### 3.5 Efficient Pathway Optimization

Given that pathway models consist of multiple layers with numerous experts per layer, optimizing all layers and experts can be computationally expensive. To mitigate this challenge, we investigate selective optimization strategies, focusing on critical layers and core experts to determine whether such targeted approaches can maintain or even enhance overall model performance. Our analysis is performed on OLMoE, optimizing only the routing weights of the last token, whose effectiveness is demonstrated in Section 4.3.

**Critical Layers** We first explore the role of critical layers by examining various layer-specific optimization strategies. Our experiments, as shown in Figure 3, systematically compare scenarios including optimization of early (F), middle (M), deep (L), and combinations of these layers. Our analysis, illustrated in Figure 3, reveals a clear hierarchy: optimizing more layers improves performance, but full-layer optimization (All16) is surprisingly inefficient. The last five layers (L5) yield the highest accuracy, outperforming both partial and full-layer optimization. This suggests that deeper layers are disproportionately responsible for refining task-specific representations, making full-layer updates computationally wasteful. Beyond the number of layers, layer positioning plays a pivotal role. A consistent pattern emerges: M1 < F1 < L1, M2 < F2 < L2, M5 < F5 < L5. Late layers contribute the most to performance, but early layers also have a greater impact than middle layers. This is likely because early layers encode fundamental feature representations, while deeper layers specialize in high-level semantic understanding. Middle layers, in contrast, appear to play a more transitional role with less direct influence on final predictions. These findings redefine optimization strategies. Instead of expending resources on full-layer updates, focusing on **critical layers—specifically, the last five—delivers superior accuracy while significantly reducing computational overhead**.

**Core Experts** After identifying the critical layers, it is also important to determine which experts within these layers should be optimized for maximum efficiency. OLMoE activates only 8 out of 64 experts per inference step for each token, making selective optimization crucial. Figure 4 illustrates the trade-off between accuracy and computational cost (FLOPs) as a function of the number of top experts (top-$n$ experts before optimization) selected for

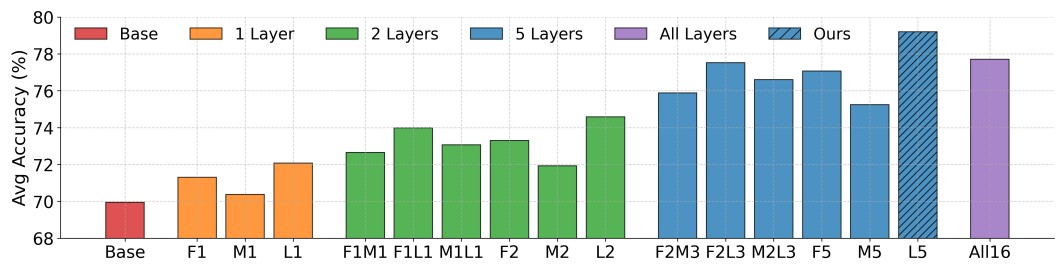

Figure 3: **Analysis of critical layers** in OLMoE (F: early layers, M: middle layers, L: late layers). Optimizing only the last five layers (L5) achieves the highest accuracy, outperforming full-layer optimization (All16) and partial combinations (e.g., F2L3).

optimization. Our experiments show that optimizing beyond the top-8 experts improves accuracy, with gains continuing up to the top-12 experts and stabilizing at the top-20. Notably, optimizing only the top-20 experts achieves the same performance as optimizing all 64, significantly reducing computational cost. Further analysis (Figure 5) reveals that optimizing the top-8 experts captures 71.3% of the final top-8 experts identified after full optimization. Expanding to the top-20 ensures 99.8% alignment, effectively covering the optimal selection. Since the top-8 activated experts (determined post-optimization) are already included in the pre-optimization top-20, peak performance is maintained with far fewer experts requiring full optimization. In summary, focusing on the **core experts**—the top-20 experts per layer—strikes an optimal balance between efficiency and accuracy, minimizing computational overhead while preserving peak performance.

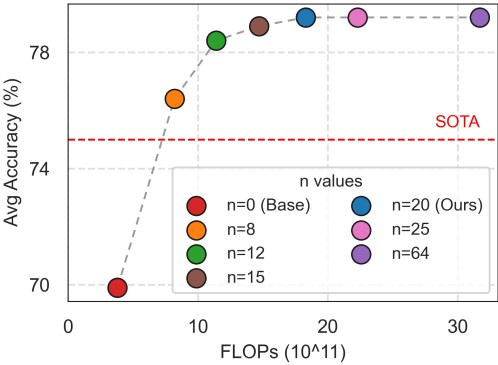

Figure 4: Accuracy-FLOPs Trade-off by changing the number of core experts ($n$) of OLMoE to optimize by C3PO. The accuracy achieves the greatest boosting at $n = 8$ and plateaus at $n = 20$, indicating **8-20 core experts suffices to retain most gain by pathway optimization.**

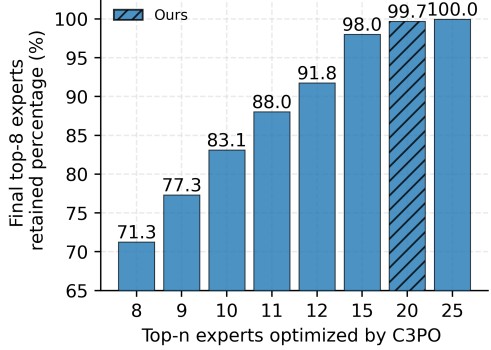

Figure 5: Average percentage of the top-8 experts (*after optimizing all experts*) being retained in the top-$n$ experts identified by pretrained router in OLMoE. The results indicate that **selecting $n \geq 20$ in advance can effectively cover almost all the 8 core experts contributing to performance.**

## 4 Experiment

### 4.1 Experimental Settings

**Models**   We evaluate two recent MoE LLMs: OLMoE and DeepSeekMoE. OLMoE uses 16 transformer layers with 64 experts per layer, activating 8 experts per token. This design yields 6.9B total parameters, with 1.3B active per token. DeepSeekMoE features a 28-layer architecture that includes 2 shared experts and 64 routed experts per layer, activating all shared experts and 6 routed experts per token. This results in 16.4B total parameters and 2.8B active parameters per forward pass.

**Evaluation benchmarks and reference sets**   We use a variety of benchmarks and reference sets across four key language model tasks. For general knowledge, we employ MMLU with BIG-Bench and SuperGLUE as references. For commonsense reasoning, we use HellaSwag and PIQA, along with CommonsenseQA and SocialIQA as references. Scientific question answering is assessed using ARC-C and ARC-E, with OpenBookQA and SciQ as references. For coreference resolution, we use WinoGrande with KnowRef as a reference. To prevent overlap, reference samples with a question similarity above 0.95 are removed during the $k$NN search. Further details are provided in Appendix A.2.

**Baselines**   We compare different variants of C3PO with both dense and MoE LLMs across various parameter scales, as shown in Tables 1 and 2. Additionally, we compare with three adaptation techniques—In-Context Learning (ICL), Prefix Tuning, and Soft Prompt Tuning. For ICL, we retrieve similar reference samples based on embedding similarity and use them as few-shot demonstrations. In contrast, Prefix Tuning and Soft Prompt Tuning are trained on the full reference sets while keeping the base model frozen.

**Evaluations**   We adopt zero-shot evaluation protocols, as our methods rely solely on external reference sets. The final performance is reported as the mean accuracy across all benchmarks.

## 4.2   Main Results

**Comparison of different baselines and C3PO methods**   Table 1 compares various methods for OLMoE and DeepSeekMoE across six benchmarks. Neighborhood Gradient Descent (NGD) consistently outperforms the base models and established baselines, achieving up to a 15.0% improvement on ARC-C for OLMoE and 10.8% for DeepSeekMoE. Although the Oracle (upper bound) represents the theoretical maximum (requiring ground truth labels at inference), NGD attains 85–95% of this potential without such labels, highlighting its effectiveness in optimizing MoE routing weights.

**Advantages of C3PO over State-of-the-Art models**   Table 2 compares LLMs across six benchmarks, categorized by active parameter counts. Notably, OLMoE-C3PO, despite using only 1B active parameters, outperforms many larger models. Among all configurations, OLMoE-C3PO delivers the best overall performance, showcasing the efficiency of our approach in maintaining competitive performance while using fewer parameters. Additional details on the baseline models can be found in Appendix A.3.

## 4.3   Ablation Study

We conduct an ablation study on OLMoE to dissect the core design choices in C3PO and their impact on performance. Specifically, we examine: (1) which tokens to optimize, (2) the effectiveness of different neighborhood selection strategies, and (3) the influence of key hyperparameters, including optimization steps and kernel function choices. Additional analyses can be found in Appendix A.7.

**Token optimization strategies**   Table 3 summarizes how routing weight optimization at different token positions affects performance in C3PO. We evaluated modifications on the first, middle, and last tokens using one or three tokens. Optimizing only the last token achieves the highest accuracy (79.20%, a 9.25% improvement over the baseline), while expanding to three tokens lowers accuracy to 77.90%. This indicates that focusing on the final token is the most effective optimization strategy.

**Neighborhood selection**   Table 4 compares neighborhood selection strategies for routing weight optimization. Both the $\epsilon$-neighborhood and $k$-Nearest Neighbors ($k$NN) methods improve upon the baseline, with $k$NN at $k = 3$ achieving the highest accuracy of 79.20% (+9.25%). Although the optimal $\epsilon$-neighborhood setting is $\epsilon = 0.5$, it still underperforms compared to $k$NN. These results suggest that a moderate number of neighbors optimally balances local adaptability and generalization.

| | MMLU | Hella-Swag | ARC-C | ARC-E | PIQA | Wino-Grande | Avg |
|---|---|---|---|---|---|---|---|
| **DeepSeekMoE** | | | | | | | |
| Base model | 46.2 | 78.0 | 50.3 | 73.8 | 79.9 | 70.1 | 66.4 |
| In-Context Learning | 49.0 | 81.6 | 56.3 | 76.2 | 81.4 | 72.3 | 69.5 |
| Prefix Tuning | 47.8 | 77.9 | 52.4 | 73.8 | 79.2 | 70.3 | 66.9 |
| Soft Prompt | 49.3 | 78.6 | 55.1 | 74.7 | 80.5 | 72.0 | 68.8 |
| Mode Finding | 48.0 | 78.8 | 57.0 | 75.9 | 81.2 | 72.0 | 68.8 |
| Kernel Regression | 53.8 | 82.3 | 59.8 | 78.9 | 84.5 | 75.8 | 72.5 |
| NGD | **55.4** | **85.7** | **61.1** | **80.7** | **85.8** | **77.5** | **74.4** |
| Oracle (upper bound) | 63.8 | 92.5 | 70.8 | 85.2 | 90.3 | 82.1 | 80.8 |
| **OLMoE** | | | | | | | |
| Base model | 57.8 | 77.9 | 51.3 | 79.8 | 80.7 | 72.2 | 69.9 |
| In-Context Learning | 60.3 | 80.6 | 58.1 | 82.5 | 83.6 | 76.8 | 73.7 |
| Prefix Tuning | 59.3 | 78.2 | 54.5 | 80.4 | 82.1 | 73.5 | 71.3 |
| Soft Prompt | 59.7 | 79.5 | 55.9 | 81.3 | 82.4 | 74.1 | 72.2 |
| Mode Finding | 58.9 | 79.1 | 57.8 | 81.8 | 82.4 | 74.3 | 72.4 |
| Kernel Regression | 63.1 | 82.0 | 64.6 | 84.7 | 86.6 | 80.2 | 76.9 |
| NGD | **65.5** | **85.3** | **66.3** | **87.4** | **88.0** | **82.7** | **79.2** |
| Oracle (upper bound) | 72.2 | 91.5 | 74.8 | 91.4 | 93.6 | 87.7 | 85.2 |

Table 1: Accuracy (%) comparison of baseline models, three C3PO variants (mode finding, kernel regression, NGD), and test-time adaptation methods (ICL, prefix tuning) across six tasks. NGD improves DeepSeekMoE by 8.0% (66.4% → 74.4%) and OLMoE by 9.3% (69.9% → 79.2%), capturing around 93% of the Oracle (upper bound).

**Step numbers** Table 5 demonstrates that the optimization step count significantly affects routing weight performance. Performance improves substantially from 3 to 10 steps (+2.5% between 3-5 steps alone), but plateaus thereafter. The minimal fluctuations at 20 and 50 steps suggest that 10 steps provide optimal balance between computational efficiency and accuracy.

**Kernel choice** Table 6 compares kernel functions for NGD. The Gaussian kernel (Williams & Rasmussen, 2006) yields the highest average accuracy (79.20%, a +9.25% improvement over the base model), outperforming the Polynomial (Cortes, 1995) (73.33%) and Matern (Williams & Rasmussen, 2006) (76.28%) kernels. This indicates that the Gaussian kernel most effectively captures non-linear relationships in high-dimensional spaces, making it optimal for routing optimization.

### 4.4 Understanding C3PO Optimization: Prediction Evolution and Expert Specialization

**Prediction Evolution: How C3PO Improves Accuracy Over Optimization Step** Figure 6 tracks the progression of predictions over 10 NGD optimization steps on ARC-C. A sharp accuracy increase (+11.6%) occurs within the first 6 steps, reaching +15.0% by Step 10. Notably, only 5.1% of initially correct predictions become incorrect, suggesting that as optimization converges, adjustments to routing weights stabilize, leading to more refined improvements rather than disruptive changes. This demonstrates the effectiveness and stability of NGD optimization in enhancing MoE model performance.

**Expert Specialization: How C3PO Refines MoE Routing** Figure 7 visualizes expert activation patterns in the last 5 layers before and after C3PO optimization. Initially, most experts remain underutilized, with only 12-20 experts being frequently activated. After

| | MMLU | Hella-Swag | ARC-C | ARC-E | PIQA | Wino-Grande | Avg |
|---|---|---|---|---|---|---|---|
| **LMs with ~1B active parameters** | | | | | | | |
| Pythia-1B | 23.1 | 45.1 | 26.2 | 48.1 | 68.7 | 52.3 | 43.9 |
| Llama3.2-1B | 27.4 | 57.9 | 32.1 | 53.9 | 72.4 | 57.4 | 50.2 |
| OLMo-1B | 24.1 | **61.8** | 29.6 | **55.7** | **75.6** | 56.8 | **50.6** |
| TinyLyne-1B-7B | **24.7** | 58.9 | **32.5** | 53.7 | 73.3 | **58.6** | 50.3 |
| **LMs with ~2-3B active parameters** | | | | | | | |
| OpenMoE-3B-9B | 23.8 | 41.5 | 25.2 | 46.3 | 59.7 | 48.2 | 40.8 |
| StableLM-2B | 31.6 | 65.1 | 37.2 | 67.2 | 76.1 | 62.6 | 56.6 |
| JetMoE-2B-9B | 39.4 | 72.6 | 51.8 | 72.1 | 73.5 | 63.4 | 62.1 |
| Gemma2-3B | 43.7 | 66.3 | 58.4 | 75.2 | 71.8 | 64.5 | 63.3 |
| Qwen1.5-3B-14B | **51.3** | **71.4** | **68.2** | **82.7** | **74.3** | **65.1** | **68.8** |
| **LMs with ~7-9B active parameters** | | | | | | | |
| Llama2-7B | 42.9 | 74.6 | 44.9 | 68.4 | 77.4 | 66.7 | 62.5 |
| Qwen-7B | 53.4 | 74.9 | 45.8 | 69.7 | 77.2 | 68.1 | 64.9 |
| Mistral-7B | 59.6 | 81.0 | 53.8 | 79.6 | 82.2 | 74.0 | 71.7 |
| DeepSeek-7B | 48.0 | 76.8 | 45.7 | 71.9 | 80.0 | 70.0 | 65.4 |
| Llama3.1-8B | 57.7 | 77.9 | 48.7 | 80.8 | 81.4 | 73.5 | 70.0 |
| OLMo2-7B | **63.2** | **85.3** | **59.7** | **83.1** | **82.3** | **76.1** | **75.0** |
| **Ours (LMs with ~1B and ~3B active parameters)** | | | | | | | |
| DeepSeekMoE-3B-16B | 46.2 | 78.0 | 50.3 | 73.8 | 79.9 | 70.1 | 66.4 |
| DeepSeekMoE-C3PO | 55.4 | 85.7 | 61.1 | 80.7 | 85.8 | 77.5 | 74.4 |
| OLMoE-1B-7B | 57.8 | 77.9 | 51.3 | 79.8 | 80.7 | 72.2 | 69.9 |
| OLMoE-C3PO | **65.5** | **85.3** | **66.3** | **87.4** | **88.0** | **82.7** | **79.2** |

Table 2: Models grouped by active parameters (1B, 2-3B, 7-9B) evaluated on six benchmarks. OLMoE-C3PO (1B active) achieves 79.2% average accuracy, outperforming most 7-9B dense models (e.g., Llama2-7B 62.5%, Mistral-7B 71.7%), demonstrating MoE+C3PO's efficiency.

| Model | Avg (%) |
|---|---|
| Base model | 69.95 |
| First 1 Token | 74.45 |
| Middle 1 Token | 71.40 |
| Last 1 Token (Ours) | **79.20** |
| First 3 Tokens | 73.63 |
| Middle 3 Tokens | 70.73 |
| Last 3 Tokens | 77.90 |

| Model | Avg (%) |
|---|---|
| Base model | 69.95 |
| $\epsilon = 0.3$ | 73.68 |
| $\epsilon = 0.5$ | 77.12 |
| $\epsilon = 0.7$ | 76.87 |
| $k = 1$ | 75.28 |
| $k = 3$ (Ours) | **79.20** |
| $k = 5$ | 77.70 |

Table 3: Optimizing pathways at token(s) of different positions (first/middle/last) and number (1 or 3 tokens) in OLMoE. Optimizing only the last token yields the best accuracy, while three-token C3PO degrades performance.

Table 4: Comparison of $\epsilon$-ball and $k$NN neighborbood in C3PO on OLMoE. $k = 3$ achieves the highest accuracy, proving moderate neighbor counts balance locality and generalization.

optimization, activation becomes more concentrated, reinforcing specialization among highly utilized experts. This suggests that C3PO refines expert selection, enabling the model to make more efficient use of a subset of core experts rather than diffusing activation across many different experts. An example of how C3PO refines MoE routing can be found in Appendix A.1.

| #Steps | Avg (%) |
|---|---|
| Base model | 69.95 |
| 3 | 74.22 |
| 5 | 76.90 |
| 10 (Ours) | 79.20 |
| 20 | **79.25** |
| 50 | 79.22 |

Table 5: Increasing NGD steps in C3PO improves the accuracy on OLMoE.

| Kernel | Avg (%) |
|---|---|
| Base model | 69.95 |
| Linear | 69.95 |
| Polynomial | 73.33 |
| Matern | 76.28 |
| Gaussian (Ours) | **79.20** |

Table 6: Comparison of different kernel choices in C3PO on OLMoE.

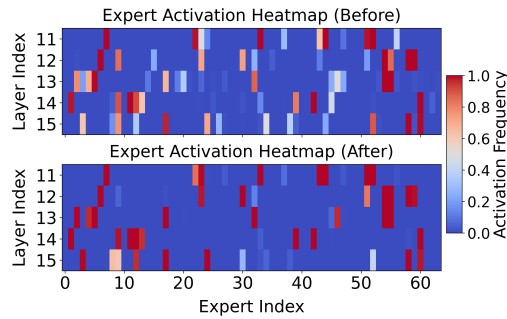

Figure 6: Impact of NGD optimization steps (x-axis) on OLMoE for ARC-C task accuracy for OLMoE. The first 6 steps yield an 11.6% gain (initial 51.3% → 62.9%), reaching 66.3% at Step 10. Only 5.1% of initially correct predictions flip, confirming stable and efficient convergence.

Figure 7: Heatmap comparison of expert activation frequency in OLMoE's last five layers for ARC-C (top: base model, right: C3PO-optimized). Post-optimization, activations concentrate, focusing on high-frequency experts per layer (darker = higher usage), showing C3PO enhances expert specialization and reduces redundancy.

## 5  Conclusions

Our work demonstrates that dynamic pathway optimization unlocks the latent potential of MoE models by addressing a critical bottleneck: suboptimal expert routing. C3PO's key insight reveals that adaptive, sample-specific routing decisions - particularly in critical layers - can significantly boost performance without architectural changes or additional training. The framework's practical impact stems from its efficient approach: by selectively optimizing only the most influential experts and layers, it achieves substantial accuracy gains while maintaining computational efficiency. This enables smaller MoE models to match or surpass larger dense counterparts, reinforcing the value of sparse architectures when properly utilized. For MoE models and beyond, dynamic adaptation of computational pathways emerges as a powerful yet underutilized strategy for improving both performance and efficiency.

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

# A  Appendix

## A.1  Example Case

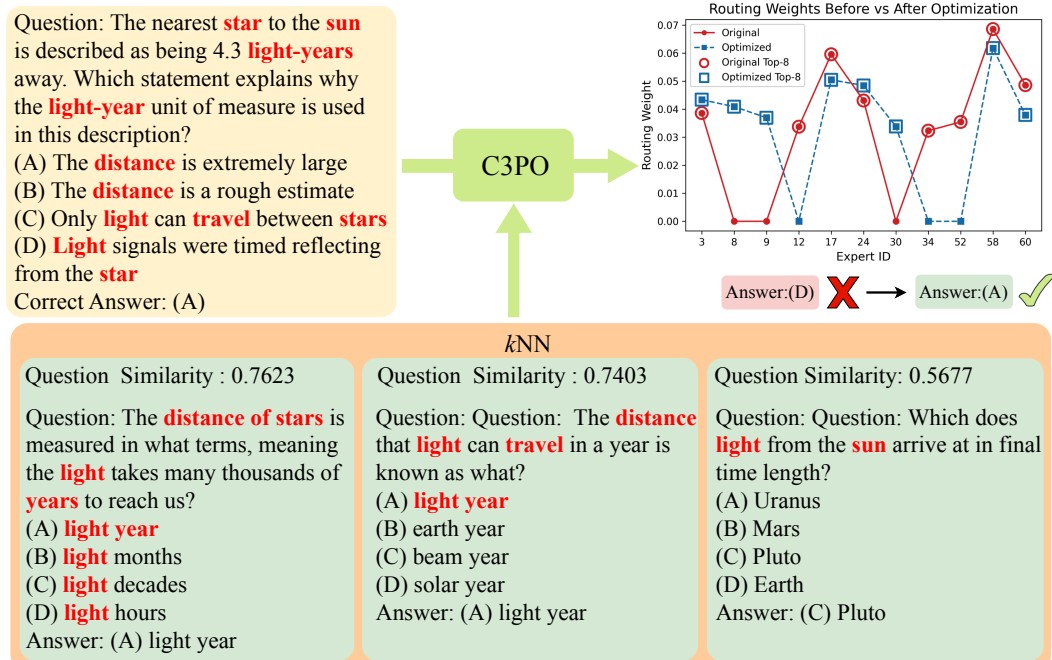

Figure 8: An example of how C3PO optimizes the expert routing weights. Here we only show the routing weights of the last1 layer. The polyline with red dots represents the original routing weights of the test sample, while the polyline with blue dots represents the optimized routing weights. C3PO optimizes the original routing weights by leveraging similar questions in the reference set, then changing the test sample's top-8 experts and their corresponding weights, eventually turning an incorrect answer into a correct one.

## A.2  Benchmarks and Reference Sets

Table 7 shows the overview of our benchmarks and reference sets.

| Task Type | Benchmarks | Size | Reference Sets | Size |
|---|---|---|---|---|
| General Knowledge | MMLU | 14,042 | BIG-Bench
SuperGLUE | 8,000
8,000 |
| Commonsense Reasoning | HellaSwag
PIQA | 10,042
1,838 | CommonsenseQA
SocialIQA | 6,000
6,000 |
| Scientific Question Answering | ARC-C
ARC-E | 1,172
2,376 | OpenBookQA
SciQ | 2,000
2,000 |
| Coreference Resolution | WinoGrande | 1,267 | KnowRef | 2,000 |

Table 7: Overview of evaluation tasks, benchmarks, and reference sets with dataset sizes.

We briefly introduce benchmarks and reference sets categorized by task types as follows:

**General Knowledge:**

- **MMLU** (Hendrycks et al., 2020): This benchmark consists of 16,000 multiple-choice questions across 57 subjects, including mathematics, philosophy, law, and medicine.

It evaluates a model's ability to understand and reason across diverse academic disciplines.

- **BIG-Bench** (Srivastava et al., 2022): A comprehensive collection of 204 tasks designed to assess the capabilities of language models beyond traditional benchmarks, covering a wide range of topics and challenges.

- **SuperGLUE** (Sarlin et al., 2020): An evolution of the GLUE benchmark, Super-GLUE comprises eight challenging language understanding tasks, including logical reasoning, commonsense inference, and coreference resolution, aimed at evaluating general language understanding.

**Commonsense Reasoning:**

- **HellaSwag** (Zellers et al., 2019): Containing 10,000 descriptions of activities or events, each with four candidate endings, this dataset challenges models to choose the most plausible continuation, testing their commonsense reasoning abilities.

- **PIQA** (Bisk et al., 2020): Comprising 17,951 two-choice questions, PIQA assesses a model's understanding of physical commonsense by evaluating its ability to choose the most effective solution to everyday tasks.

- **CommonsenseQA** (Talmor et al., 2018): A dataset with 12,102 multiple-choice questions that require models to utilize commonsense knowledge to select the correct answer, focusing on everyday scenarios and concepts.

- **SocialIQA** (Sap et al., 2019): Featuring 38,000 multiple-choice questions, SocialIQA evaluates a model's understanding of social interactions and norms by assessing its ability to reason about social situations and their implications.

**Scientific Question Answering:**

- **ARC-C** (Clark et al., 2018): Consisting of 2,590 multiple-choice science questions, the Challenge Set is designed to be difficult for state-of-the-art models, requiring advanced reasoning and knowledge.

- **ARC-E** (Clark et al., 2018): With 5,197 multiple-choice science questions, the Easy Set serves as a baseline to evaluate a model's performance on straightforward scientific queries.

- **OpenBookQA** (Mihaylov et al., 2018): This dataset includes 5,957 multiple-choice questions, each associated with an elementary science fact (the "open book"), assessing a model's ability to apply core scientific principles to answer questions.

- **SciQ** (Welbl et al., 2017): Containing 13,679 science questions, SciQ is designed to evaluate a model's proficiency in answering questions across various scientific domains, including biology, chemistry, and physics.

**Coreference Resolution:**

- **WinoGrande** (Sakaguchi et al., 2021): An expanded version of the Winograd Schema Challenge, WinoGrande comprises 44,000 fill-in-the-blank style sentences that test a model's ability to resolve ambiguous pronouns within diverse contexts.

- **KnowRef** (Emami et al., 2018): This dataset contains 8,311 sentences with ambiguous pronouns, challenging models to accurately determine the antecedents of pronouns in complex sentences.

## A.3 Baseline Models

We briefly introduce each model categorized by active parameter size as follows:

**LMs with ~1B active parameters:**

- **Pythia-1B** (Biderman et al., 2023): A 1-billion-parameter dense model, trained by EleutherAI using standard autoregressive training techniques.

- **Llama3.2-1B** (Grattafiori et al., 2024): A compact variant of the Llama family, featuring approximately 1 billion parameters designed by Meta.

- **OLMo-1B** (Groeneveld et al., 2024): An open-source dense transformer model with around 1 billion parameters, developed by Allen Institute for AI (AI2).

- **TinyLyne-1B-7B** Tang et al. (2024): A sparse mixture-of-experts (MoE) model with 1 billion active parameters from a total of 7 billion parameters.

**LMs with ~2-3B active parameters:**

- **OpenMoE-3B-9B** (Xue et al., 2024): An MoE architecture having 3 billion active parameters selected from a total of 9 billion parameters.

- **StableLM-2B** (Bellagente et al., 2024): A dense transformer-based language model by Stability AI, containing around 2 billion parameters.

- **JetMoE-2B-9B** (Shen et al., 2024): A sparse mixture-of-experts model from the Jet series with 2 billion active parameters chosen from a pool of 9 billion.

- **Gemma2-3B** (Team et al., 2024): A dense transformer model developed by Google DeepMind with approximately 3 billion parameters.

- **Qwen1.5-3B-14B** (Yang et al., 2024): A large-scale MoE model by Alibaba, featuring 3 billion active parameters selected from a total of 14 billion parameters.

**LMs with ~7-9B active parameters:**

- **Llama2-7B** (Touvron et al., 2023): Meta's open-source dense language model with approximately 7 billion parameters.

- **Qwen-7B** (Yang et al., 2024): A 7-billion-parameter dense transformer model developed by Alibaba.

- **Mistral-7B** (Jiang et al., 2023): A dense language model by Mistral AI, consisting of roughly 7 billion parameters.

- **DeepSeek-7B** (Bi et al., 2024): An open-source transformer-based dense language model with 7 billion parameters.

- **Llama3.1-8B** (Grattafiori et al., 2024): Meta's latest generation dense transformer model with about 8 billion parameters.

- **OLMo2-7B** (OLMo et al., 2024): An advanced 7-billion-parameter dense model by Allen Institute for AI, building upon the OLMo architecture.

### A.4 Impact of Reference Set Size

We study how the size of the reference set affects C3PO's effectiveness. We systematically varied the reference set size from 20 to ~34K examples (randomly sampled) and evaluated C3PO's improvement over OLMoE. With very small pools (<1K), performance gains were negligible (±0.4%). However, once the size reached 10K, accuracy improved significantly (+5.0%) and continued to increase gradually, reaching +9.3% at ~34K. This demonstrates that larger reference sets substantially boost C3PO's performance, with diminishing returns beyond ~10K (29%).

### A.5 Inference Overhead vs. Accuracy

We quantify the real-world inference cost of C3PO beyond theoretical FLOPs. Using the default configuration (10 NGD steps, 3 neighbors), per-sample latency increases from 1.8s (baseline) to 5.1s (≈2.8×), while average accuracy improves from 69.9% to 79.2%. For lower-latency regimes, smaller configurations still yield meaningful gains (e.g., 5 steps with 3 neighbors achieves 76.9% accuracy at 3.2s). Embedding storage for the full reference set remains lightweight (~20 MB in Parquet), making the method practically deployable with a controllable accuracy/latency trade-off.

| Reference Size | Avg Accuracy (%) | vs. Baseline (%) | vs. Previous size (%) |
|---|---|---|---|
| Baseline | 69.9 | 0.0 | – |
| 20 | 70.0 | +0.1 | +0.1 |
| 200 | 69.8 | -0.1 | -0.2 |
| 1,000 | 70.3 | +0.4 | +0.5 |
| 10,000 (29%) | 74.9 | +5.0 | +4.6 |
| 17,000 (50%) | 76.7 | +6.8 | +1.8 |
| 27,000 (80%) | 78.3 | +8.4 | +1.6 |
| Full (∼34K) | 79.2 | +9.3 | +0.9 |

Table 8: Effect of reference set size on OLMoE-C3PO.

| Steps | Neighbors | Avg Accuracy (%) | Latency per Sample (s) |
|---|---|---|---|
| Baseline (1, 0) | 0 | 69.9 | 1.8 |
| 5 | 1 | 72.6 | 2.7 |
| 5 | 3 | 76.9 | 3.2 |
| 5 | 5 | 74.7 | 3.4 |
| 10 | 1 | 75.3 | 4.5 |
| 10 (Ours) | 3 | 79.2 | 5.1 |
| 10 | 5 | 77.7 | 5.5 |

Table 9: Accuracy vs. real-time inference latency for different C3PO configurations.

## A.6 Pathway Evolution and Expert Redistribution

We analyze how C3PO modifies expert activations to improve predictions. Focusing on the final layer for ARC-C, C3PO reduces over-reliance on some initially dominant experts (e.g., expert9 and expert52 decrease by 7.2% and 9.0%, respectively) while boosting more informative ones (expert12 and expert17 increase by 13.1% and 6.0%). This redistribution correlates with the accuracy improvement from 51.3% to 66.3%. Furthermore, we perform interpolation between the base and optimized routing weights, observing that as the interpolation coefficient $\alpha$ increases, both the concentration on top experts and accuracy steadily improve (see Table 10), confirming that C3PO sharpens specialization rather than indiscriminately expanding activation.

| Interpolation $\alpha$ | Top-8 Activation Frequency | Avg Accuracy (%) |
|---|---|---|
| 0.0 (baseline) | 0.65 | 69.9 |
| 0.3 | 0.71 | 72.7 |
| 0.5 | 0.76 | 74.6 |
| 0.7 | 0.79 | 76.4 |
| 1.0 (C3PO) | 0.83 | 79.2 |

Table 10: Effect of interpolating between original and C3PO-optimized pathways on expert concentration and accuracy.

## A.7 Ablation Study

**Layer optimization strategies**   determine which specific layers' routing weights should be modified in each token, directly influencing the model's performance after optimization. Table 11 analyzes different layer optimization strategies for routing weights in OLMoE. We systematically explore various combinations within the OLMoE's 16 layers, revealing that the location of optimized layers significantly impacts performance. Single-layer optimization shows best results when targeting the last layer, while two-layer combinations including the last layer consistently outperform other configurations. Most importantly,

| OLMoE | MMLU | HellaSwag | ARC-C | ARC-E | PIQA | WinoGrande |
|---|---|---|---|---|---|---|
| Base model | 57.8 | 77.9 | 51.3 | 79.8 | 80.7 | 72.2 |
| **1 Layer Optimization** | | | | | | |
| First 1 | 59.4 | 78.9 | 52.8 | 80.3 | 82.5 | 73.9 |
| Middle 1 | 58.3 | 78.1 | 51.9 | 79.9 | 81.2 | 72.8 |
| Last 1 | 60.2 | 79.7 | 53.5 | 81.6 | 82.9 | 74.5 |
| **2 Layers Routing Weights Optimization** | | | | | | |
| First 1 + Middle 1 | 60.5 | 80.2 | 54.6 | 82.3 | 83.1 | 75.2 |
| First 1 + Last 1 | 61.8 | 81.3 | 55.8 | 83.7 | 84.5 | 76.8 |
| Middle 1 + Last 1 | 60.9 | 80.7 | 54.9 | 82.8 | 83.4 | 75.7 |
| First 2 | 60.7 | 80.6 | 55.3 | 83.1 | 84.0 | 76.1 |
| Middle 2 | 59.9 | 79.5 | 53.9 | 81.9 | 82.3 | 74.1 |
| Last 2 | 62.3 | 81.9 | 56.7 | 84.2 | 85.1 | 77.3 |
| **5 Layers Routing Weights Optimization** | | | | | | |
| First 2 + Middle 3 | 63.2 | 82.8 | 59.4 | 85.1 | 85.6 | 79.2 |
| First 2 + Last 3 | 64.3 | 83.7 | 62.8 | 86.5 | 87.1 | 80.7 |
| Middle 2 + Last 3 | 63.7 | 83.1 | 61.5 | 85.3 | 86.2 | 79.8 |
| First 5 | 63.9 | 83.5 | 62.1 | 85.9 | 86.7 | 80.3 |
| Middle 5 | 62.5 | 82.3 | 58.7 | 84.6 | 84.9 | 78.5 |
| Last 5 | **65.5** | **85.3** | **66.3** | **87.4** | **88.0** | **82.7** |
| **All Layers Routing Weights Optimization** | | | | | | |
| All (16) Layers | 64.1 | 84.3 | 63.7 | 86.1 | 86.8 | 81.2 |

Table 11: Comparison of C3PO applied to different layers in OLMoE. Performance comparison of different layer optimization strategies.

optimizing only the final five layers (Last5) achieves the best performance across all benchmarks, surpassing even the full 16-layer optimization (All16). This suggests that focusing optimization on the deeper layers near the output is more effective than modifying the entire network, highlighting the importance of targeted layer selection in MoE architectures.

**Token optimization strategies** determine which specific token numbers and positions should be modified in the sequence, significantly affecting the inference results after optimization. Table 12 examines the impact of optimizing routing weights at different token positions in OLMoE. We systematically analyze various positions (first, middle, last) and quantities (one, three tokens). Results clearly show that token position significantly affects performance, with last token optimization consistently outperforming other configurations across all benchmarks. Notably, optimizing only the last token yields the best results, achieving improvements of +7.7% on MMLU and +15.0% on ARC-C compared to the baseline. Expanding optimization to three tokens actually decreases performance, suggesting that focusing exclusively on the final token provides the most effective routing optimization strategy.

**Neighborhood selection** Table 13 examines different neighborhood selection strategies for routing weight optimization in OLMoE. We evaluate two approaches: an $\epsilon$-neighborhood method with various thresholds and a $k$-nearest neighbors ($k$NN) approach with different $k$ values. While both methods significantly improve performance over the baseline, the $k$NN approach with $k$=3 consistently delivers the best results across all benchmarks, achieving improvements of +7.7% on MMLU and +15.0% on ARC-C. The $\epsilon$-neighborhood method shows strong performance at $\epsilon$=0.5, but still falls short of $k$NN's effectiveness. These results indicate that selecting a moderate number of nearest neighbors provides the optimal strategy for neighborhood-based routing optimization.

| | MMLU | HellaSwag | ARC-C | ARC-E | PIQA | WinoGrande |
|---|---|---|---|---|---|---|
| Base model | 57.8 | 77.9 | 51.3 | 79.8 | 80.7 | 72.2 |
| **1 Token Optimization** | | | | | | |
| First 1 Token | 61.4 | 81.5 | 58.7 | 83.6 | 84.2 | 77.3 |
| Middle 1 Token | 59.2 | 79.1 | 53.0 | 81.2 | 82.1 | 73.8 |
| Last 1 Token | **65.5** | **85.3** | **66.3** | **87.4** | **88.0** | **82.7** |
| **3 Tokens Optimization** | | | | | | |
| First 3 Token | 60.8 | 80.7 | 57.5 | 82.9 | 83.5 | 76.4 |
| Middle 3 Token | 58.6 | 78.5 | 52.4 | 80.5 | 81.3 | 73.1 |
| Last 3 Token | 64.1 | 84.3 | 64.8 | 86.2 | 86.7 | 81.3 |

Table 12: Performance comparison of different token optimization strategies.

| | MMLU | HellaSwag | ARC-C | ARC-E | PIQA | WinoGrande |
|---|---|---|---|---|---|---|
| Base model | 57.8 | 77.9 | 51.3 | 79.8 | 80.7 | 72.2 |
| $\epsilon = 0.3$ | 60.4 | 80.5 | 57.2 | 83.4 | 84.1 | 76.5 |
| $\epsilon = 0.5$ | 63.2 | 83.7 | 63.5 | 85.8 | 86.3 | 80.2 |
| $\epsilon = 0.7$ | 62.8 | 84.1 | 62.9 | 85.1 | 86.5 | 79.8 |
| $k = 1$ | 61.7 | 82.3 | 59.8 | 84.2 | 85.3 | 78.4 |
| $k = 3$ (Ours) | **65.5** | **85.3** | **66.3** | **87.4** | **88.0** | **82.7** |
| $k = 5$ | 63.9 | 84.5 | 63.7 | 86.1 | 86.7 | 81.3 |

Table 13: Performance comparison of different optimization strategies.

**Step numbers** Table 14 examines how the number of optimization steps affects routing weight performance in OLMoE. Results show significant improvements as steps increase from 3 to 10, with substantial early gains (+2.5% on MMLU from 3 to 5 steps) that gradually diminish due to our cosine annealing learning rate schedule. Importantly, performance plateaus beyond 10 steps, with minimal fluctuations at 20 and 50 steps across all benchmarks. This indicates that 10 optimization steps provide a better balance between computational efficiency and performance improvement, as additional steps yield negligible benefits.

| #Steps | MMLU | HellaSwag | ARC-C | ARC-E | PIQA | WinoGrande |
|---|---|---|---|---|---|---|
| Base model | 57.8 | 77.9 | 51.3 | 79.8 | 80.7 | 72.2 |
| 3 | 61.3 | 81.2 | 58.3 | 83.3 | 84.0 | 77.2 |
| 5 | 63.8 | 83.4 | 62.5 | 85.4 | 86.2 | 80.1 |
| 7 | 64.8 | 84.7 | 65.2 | 86.8 | 87.3 | 81.7 |
| 10 (Ours) | 65.5 | 85.3 | 66.3 | 87.4 | 88.0 | 82.7 |
| 20 | 65.4 | 85.7 | 66.5 | 87.2 | 88.3 | 82.4 |
| 50 | 65.7 | 85.2 | 66.1 | 87.5 | 87.9 | 82.9 |

Table 14: Performance comparison with different numbers of optimization steps.

**Learning rate** Table 15 demonstrates the impact of learning rate schedules on model performance across six benchmarks. The cosine learning rate schedule (10e-2 → 10e-5) consistently outperforms other methods, achieving improvements of +7.7% on MMLU, +7.4% on HellaSwag, and +15.0% on ARC-C over the base model. Step decay (10e-2 → 10e-5) shows comparable but slightly lower gains, while fixed learning rates (1e-4 and 1e-3) yield more modest improvements. These results highlight that adaptive learning rate strategies, particularly cosine scheduling, significantly enhance model performance.

| Learning Rate | MMLU | HellaSwag | ARC-C | ARC-E | PIQA | WinoGrande |
|---|---|---|---|---|---|---|
| Base model | 57.8 | 77.9 | 51.3 | 79.8 | 80.7 | 72.2 |
| Fixed(1e-3) | 59.1 | 79.4 | 53.0 | 81.2 | 82.1 | 73.9 |
| Fixed(1e-4) | 61.5 | 81.6 | 57.1 | 83.5 | 84.3 | 76.8 |
| Step Decay | 64.8 | 84.7 | 65.3 | 86.8 | 87.2 | 81.9 |
| Cosine(Ours) | **65.5** | **85.3** | **66.3** | **87.4** | **88.0** | **82.7** |

Table 15: Performance comparison with different learning rate schedules.

**Embedding model**  Table 16 demonstrates the significant impact of embedding model quality on performance across six benchmarks. NV-Embed-V2 consistently outperforms other embedding models, achieving improvements of up to +15.0% on ARC-C compared to the base model. The results show the clear improvement from All-Mini-V6 to our NV-Embed-V2. This trend confirms that higher-quality embeddings enable more effective identification of relevant neighbors in the reference set, which directly translates to better optimization of routing weights and enhanced performance on downstream tasks.

| Embedding Model | MMLU | HellaSwag | ARC-C | ARC-E | PIQA | WinoGrande |
|---|---|---|---|---|---|---|
| Base model | 57.8 | 77.9 | 51.3 | 79.8 | 80.7 | 72.2 |
| All-Mini-V6 | 58.9 | 78.6 | 53.5 | 80.3 | 82.3 | 73.9 |
| Sentence-Bert | 61.2 | 80.8 | 56.1 | 83.7 | 83.1 | 77.4 |
| Stella-En-1.5B-V5 | 62.1 | 83.4 | 61.2 | 84.2 | 85.8 | 78.3 |
| Gte-Qwen2-7B-instruct | 64.5 | 83.9 | 62.8 | 86.5 | 85.2 | 81.4 |
| NV-Embed-V2 (Ours) | **65.5** | **85.3** | **66.3** | **87.4** | **88.0** | **82.7** |

Table 16: Performance comparison with different embedding models.

**Kernel choice**  Table 17 compares different kernel functions for NGD across six benchmarks. The Gaussian kernel consistently outperforms alternatives, achieving substantial improvements over the linear baseline (+7.7% on MMLU, +7.4% on HellaSwag, +15.0% on ARC-C). This result suggests the Gaussian kernel's effectiveness stems from its superior ability to model non-linear relationships in high-dimensional embedding spaces.

| Kernel | MMLU | HellaSwag | ARC-C | ARC-E | PIQA | WinoGrande |
|---|---|---|---|---|---|---|
| Linear | 57.8 | 77.9 | 51.3 | 79.8 | 80.7 | 72.2 |
| Polynomial | 61.2 | 79.4 | 58.7 | 81.5 | 82.9 | 76.3 |
| Matern | 62.9 | 83.1 | 61.8 | 85.2 | 84.5 | 80.2 |
| Gaussian (Ours) | **65.5** | **85.3** | **66.3** | **87.4** | **88.0** | **82.7** |

Table 17: Performance comparison with different kernel functions.

