# OpenReview forum: "C3PO: Critical-Layer, Core-Expert, Collaborative Pathway Optimization for Test-Time Expert Re-Mixing"
_colmweb.org/COLM/2025/Conference — COLM 2025_

### Official Review · Reviewer_Qu61 · 2025-05-02

**Rating:** 6
**Confidence:** 3
**Ethics Flag:** 1

**Summary:**

This paper presents C3PO to optimize the inference-time token-wise expert routing in the MoE system according to the routing paths obtained from similar instances in a reference set that lead to the correct results. Experiments show that the proposed method effectively improves the model performance on multiple reasoning datasets when accompanied by strong and relevant reference sets. The proposed method could be impactful and inspiring to the community if some issues is properly addressed.

**Questions To Authors:**

I might have missed it, but how is function $d$ introduced in Line 144 defined?

In Lines 197--198, the authors mention that "reference samples with a question similarity above 0.95 are removed". How is the similarity calculated? Is the threshold set too high (I was expecting something like 0.5 or 0.3)? How will this threshold impact the results?

I don't fully understand the discussion in Lines 253--259. From Figure 7, it looks like more experts are used in the base model than C3PO. Are you saying it is a good thing or a bad thing? Why? Can you prove your opinion?

**Reasons To Accept:**

The article is overall well-written. The proposed method has a strong performance under the proposed setup. The authors have done a set of ablation studies to verify the effectiveness of their design.

**Reasons To Reject:**

I feel that the applicability of C3PO might be a big issue. According to the paper, the authors use very strong and relatively large and diverse reference sets in their experiments, which casts doubt on whether the proposed method will still work under the practical situation, where the reference set is weaker (easier to answer) than the test set, and its size is more limited, say, less than 100 instances. Therefore, I recommend the following experiments:
- Use GSM8k or SWAG as reference, and MATH or harder mathematical reasoning datasets as test.
- Study the impact of reference instance number (ranges from ~20 to all).
- Compare C3PO to a model fine-tuned (with SFT or DPO or other RL methods) on the reference set.

In addition, the proposed method introduces inference overhead, which may further hinder its applicability. The authors are advised to study this issue with experiments.

The authors could have used a pipeline figure or a pseudo-code block to better illustrate the proposed C3PO pipeline.

---

> ### Author Response · Authors · 2025-06-03
> **Response to Reviewer Qu61 (2/2)**
>
> ## Response to Reviewer Qu61
>
> We sincerely appreciate the reviewer's comprehensive assessment and constructive suggestions that highlighted areas needing further clarification. We address your comments below.
>
> >**Q5: The authors could have used a pipeline figure or a pseudo-code block to better illustrate the proposed C3PO pipeline**
>
> We appreciate the suggestion. In our work, the C3PO workflow is already illustrated through two complementary figures:
>
> - Figure2 shows the overview of C3PO’s pathway optimization. For an input sample, C3PO finds a reference case in the Reference Set whose question embedding is similar to the input, obtaining a successful pathway. Finally, C3PO combines the original pathway and the successful pathway to produce an optimized pathway, resulting in improved model predictions.
>
> - Figure8 provides a detailed example of how C3PO optimizes the expert routing weights. C3PO finds a similar question in the reference set based on embedding similarity; These questions share many of the same keywords noted in the red color. After C3PO optimization, we observe that the routing weights for the top-8 experts in the final layer have nontrivially changed, causing the model’s prediction to shift from incorrect to correct.
>
>
> >**Q6: I might have missed it, but how is function $d$ introduced in Line 144 defined?**
>
> We apologize for the confusion. In our paper, $d(x, x_i)$ is defined as one minus the cosine similarity between the embedding vectors of $x$ and $x_i$:
>
>
> $$
> d(x, x_i) = 1 - \cos\bigl(E(x), E(x_i)\bigr).
> $$
>
> We will explicitly provide this definition in the paper.
>
>
> >**Q7: In Lines 197--198, the authors mention that "reference samples with a question similarity above 0.95 are removed". How is the similarity calculated? Is the threshold set too high (I was expecting something like 0.5 or 0.3)? How will this threshold impact the results?**
>
> We calculate question similarity using cosine similarity between question embeddings. The 0.95 threshold ensures removal of near-duplicate questions between the test and reference sets, effectively preventing test-case leakage. Setting a lower threshold, such as 0.5 or 0.3, would incorrectly exclude many distinct reference questions that merely share a few common keywords or topics, significantly reducing the reference set’s diversity and coverage. Therefore, the 0.95 threshold strikes a crucial balance: it eliminates genuine overlap while retaining a rich and diverse set of useful reference questions. We compare different threhold settings in the table below. When the threshold set to 0.3, the reference set size drops to ≈11K and average accuracy falls to 73.3% (a 6% decrease from 79.2% at 0.95 threshold). At 0.5, we retain ≈18K questions but only achieve 75.5% accuracy, 3.7% lower than 79.2% using 34K references at 0.95 threshold.
>
>
> | Threshold     | Reference Set Size | Avg. Accuracy (%)    |
> | ------------- | ------------------ | -------------------- |
> | Baseline      | -                  | 69.9                 |
> | 0.3           | ~11K               | 73.3 (+3.4)          |
> | 0.5           | ~18K               | 75.5 (+5.6)          |
> | 0.95 (Ours)   | 34K                | 79.2 (+9.3)          |
>
>
>
> >**Q8: I don't fully understand the discussion in Lines 253--259. From Figure 7, it looks like more experts are used in the base model than C3PO. Are you saying it is a good thing or a bad thing? Why? Can you prove your opinion?**
>
> Thank you for the question. The key idea is that C3PO deliberately concentrates on a smaller subset of highly relevant experts, rather than diluting attention across many experts. Although the base model may activate more experts overall (as shown in Figure 7), those additional experts often contribute noisy or irrelevant information. In contrast, C3PO optimized routing weights concentrate on the most relevant experts for each token, enhancing prediction accuracy.
>
> To support this claim, we conducted an interpolation experiment between the base model’s original routing weights (α=0) and the C3PO-optimized weights (α=1).
>
> | Interpolation α | Avg. Activated Frequency for Top-8 Expert | Avg. Accuracy (%) |
> | :-------------: | :---------------------------------------: | :---------------: |
> | 0 (Baseline)   |                    0.65                   |      69.9 (-)     |
> |       0.3      |                    0.71                   |    72.7 (+2.8)    |
> |       0.5      |                    0.76                   |    74.6 (+4.7)    |
> |       0.7      |                    0.79                   |    76.4 (+6.5)    |
> |   1.0 (Ours)   |                    0.83                   |    79.2 (+9.3)    |
>
> The results show that as α increases, the model’s accuracy improves, reaching a maximum of 79.2 when α=1.0 (C3PO). At the same time, the activation frequency of the top-8 experts also increases with α, demonstrating that the activated experts become increasingly concentrated in a small subset of highly relevant experts.

---

> > ### Comment · Reviewer_Qu61 · 2025-06-06
> >
> > Thanks for your response. It addresses my concerns. Please include a more formal version of the discussion into the final revision if the paper is accepted.

---

> > > ### Author Response · Authors · 2025-06-06
> > > **Response to Reviewer Qu61**
> > >
> > > Thank you for your feedback! We appreciate your suggestion and will be sure to incorporate a more formal version of the discussion into the final revision.

---

> ### Author Response · Authors · 2025-06-03
> **Response to Reviewer Qu61(1/2)**
>
> ## Response to Reviewer Qu61
>
> We sincerely appreciate the reviewer's comprehensive assessment and constructive suggestions that highlighted areas needing further clarification. We address your comments below.
>
> > **Q1: Use GSM8k or SWAG as reference, and MATH or harder mathematical reasoning datasets as test.**
>
> Thank you for your question. We evaluated C3PO using GSM8K as the reference set and MATH (~12.5K problems) as the test set, and compared its performance with the base model OLMoE (21.4% accuracy).
>
> | Reference Set | Reference Set Size | Test Set (~12.5K) | C3PO Accuracy (%) | OLMoE Accuracy (%) | Improvement (%) |
> | ------------: | -----------------: | -----------------: | -----------------: | --------------------: | -----------------------: |
> |         GSM8K |                100 | MATH              | 21.6              | 21.4                 | +0.2                    |
> |         GSM8K |               1000 | MATH              | 23.1             | 21.4                 | +1.7                 |
> |         GSM8K |               4000 | MATH              | 24.4           | 21.4                 | +3.0        |
> |         GSM8K |       Full (~8.8K) | MATH              | 26.2        | 21.4                 | +4.8          |
>
> These results show that C3PO’s benefits over OLMoE are minimal with very small reference sets but become substantial once the reference pool exceeds a few thousand examples.
>
> >**Q2: Study the impact of reference instance number (ranges from ~20 to all)**
>
> Please see our General Response on Reference Set Size for a detailed discussion supported by a table of new results.
>
>
> >**Q3: Compare C3PO to a model fine-tuned (with SFT or DPO or other RL methods) on the reference set**
>
> Thank you for your question. We fine-tuned the same OLMoE backbone on the reference set using SFT and DPO, and then compared their test accuracies to C3PO. SFT achieved 71.5% (+1.6% over baseline), and DPO reached 73.4% (+3.4%). In contrast, C3PO achieved 79.2% (+9.3%), showing that its test-time pathway optimization brings substantially more improvements in aligning with human-preferred answers than full-model finetuning such as SFT or DPO.
>
> | Method           | Accuracy on Test Set (%) | Improvement (%) |
> | ---------------- | ------------------------ | --------------- |
> | Baseline (OLMoE) | 69.9                     | 0               |
> | C3PO             | 79.2                     | +9.3            |
> | SFT              | 71.5                     | +1.6            |
> | DPO              | 73.4                     | +3.4            |
>
>
> >**Q4: the proposed method introduces inference overhead, which may further hinder its applicability. The authors are advised to study this issue with experiments**
>
> Please see our General Response for detailed experiments and discussion of this trade-off.

---

### Official Review · Reviewer_hK3h · 2025-05-10

**Rating:** 7
**Confidence:** 4
**Ethics Flag:** 1

**Summary:**

This paper explores the important problem of test-time adaptation in MoE models. The authors demonstrate that the default expert routing is often suboptimal, and that downstream performance can be significantly improved by optimizing the routing pathways at inference time. They evaluate three distinct strategies for this purpose. The proposed method is promising and highlights a valuable yet underexplored direction in scaling MoE LLMs during test time.

**Questions To Authors:**

**Increased test-time compute?** Can the authors quantify the increased test-time compute and storage (for the embeddings of the reference set) for “re-mixing” the experts? As the pathway optimization is per-sample based, is it costly to do such optimization? It would be nice to present the trade-off between accuracy and test-time compute.

**Reasons To Accept:**

- The paper is well-written and the results are clearly and nicely presented.

- The paper proposes a simple yet effective approach for test-time MoE optimization, and a comprehensive ablation study is provided.

**Reasons To Reject:**

**The requirement of reference set might not be realistic.** The reference set is task-specific, which limits the usage of the proposed method, as sometimes such a reference set might not be available. Could you further provide details on what the minimal requested size for such a reference set is?

---

> ### Author Response · Authors · 2025-06-03
> **Response to Reviewer hK3h**
>
> ## Response to Reviewer hK3h
>
> We appreciate your thorough review and insightful comments that helped us identify areas needing clarification. We address your comments below.
>
> > **Q1: Could you further provide details on what the minimal requested size for such a reference set is?**
>
> Please see our General Response on Reference Set Size for a detailed discussion and table of results.
>
>
> > **Q2: Can the authors quantify the increased test-time compute and storage (for the embeddings of the reference set) for “re-mixing” the experts? As the pathway optimization is per-sample based, is it costly to do such optimization? It would be nice to present the trade-off between accuracy and test-time compute.**
>
> Please see our General Response, which provides a detailed discussion and trade-off analysis.

---

> > ### Comment · Reviewer_hK3h · 2025-06-06
> >
> > Thanks for your response. It does seem that the proposed method requires a fairly large reference set, which, in my view, may limit its broader applicability. I will keep my score as it is.

---

> > > ### Author Response · Authors · 2025-06-06
> > > **Response to Reviewer hK3h**
> > >
> > > Thank you for your continued feedback! There exist many techniques (like FAISS) that can substantially accelerate retrieval and mitigate this limitation. In fact, retrieval‐augmented methods like RAG face a similar challenge, and the same acceleration strategies apply. Nevertheless, our main focus in this work is on test‐time adaptation, rather than retrieval speed. Thank you again for pointing this out!

---

### Official Review · Reviewer_7Z5F · 2025-05-14

**Rating:** 7
**Confidence:** 4
**Ethics Flag:** 1

**Summary:**

This paper proposes a test-time optimization method for Mixture-of-Experts (MoE) models that tunes the expert selection mechanism. Compared to existing test-time optimization methods such as prompt tuning, the number of parameters that need to be learned is smaller, and it is possible to apply optimization methods other than gradient-based optimization, which is a key characteristic. The paper validates two types of such optimization methods, together with another one requiring gradient-based optimization. It also demonstrates that performance improvement can be expected even by limiting the optimization target to only a small subset (last 5 layers chosen according to the ablation) rather than optimizing the entire network pathway. The proposed method, C3PO, which summarizes these findings, is shown to significantly improve model performance compared to existing methods.

**Questions To Authors:**

* The scales of the axes in Figure 1 are completely different, which clearly misleads the reader. Correcting this figure is essential.
* Since the reference set and evaluation benchmarks are the same, the setup might be prone to yielding higher scores. While it seems correct that this method is suited for extreme optimization on specific tasks, more discussion is needed regarding its effectiveness in terms of generalization performance across a wider range of tasks. Also, to be sure, could you explicitly state that there is no possibility of test leakage due to this method?
* The concept of "successful" used in the paper seems ambiguous. What quantitative criteria are used? Is it simply samples that were answered correctly?
* Can you discuss the relationship between the size of the reference set and the degree of performance improvement? In practice, there is a possibility that a proprietary reference set must be prepared to apply this method, so this estimate is important from a practical standpoint.
* The ablation study seems to use only OLMoE. Are the hyperparameters found here applicable to other models without issues?
* Similarly, how generalizable is the conclusion about critical layers shown in Section 3.5 to models other than OLMoE?
* We would like a more concrete discussion on how the pathway changes actually caused by C3PO affect the performance.
* How generally applicable are C3PO or similar strategies to routing algorithms other than those used in this paper? While it feels like a method that could be used quite generally with slight modifications, are there any model structures where the method would not be applicable?
* What is the real-time delay of this method? FLOPs are often theoretical and do not reflect the computational efficiency of actual hardware architectures. Is it possible to calculate inference cost in seconds along with FLOPs?
* In building the reference set, how do you handle samples that include incorrect predictions? For example, is there a possibility that errors in the reference set could negatively impact the optimization? We are interested in the degree of negative impact (i.e., how much effort is needed to build a clean reference).

**Reasons To Accept:**

* The paper focuses on the importance of routing in MoE models and demonstrates that significant performance improvement can be achieved with high efficiency by optimizing it.
* By adopting a test-time optimization approach rather than modifying the model itself, the required computation can be kept low.
* The approximate solution using routing from reference examples is clever and might potentially be applicable to other test-time optimization methods.
* Based on the experimental results, the performance improvement offered by the proposed method is highly promising.

**Reasons To Reject:**

* The theoretical grounding for the surrogate objective is weak, and the explanation seems limited to stating that the proposed method works empirically. For example, there appear to be quite strong assumptions about the similarity between successful neighbors and pathway similarity. While empirical success is important, the paper needs more explanation on these aspects. Also, the quantitative discussion on the characteristics of each surrogate objective seems insufficient.
* The method requires accepting newly introduced models and their associated hyperparameters as new elements, which poses a potential issue for portability compared to using a single pretrained model.

---

> ### Author Response · Authors · 2025-06-03
> **Response to Reviewer  7Z5F (2/2)**
>
> ## Response to Reviewer  7Z5F
>
> Thank you for your detailed feedback! We address your comments below.
>
> > **Q7: The ablation study seems to use only OLMoE. Are the hyperparameters found here applicable to other models without issues?**
>
> Thank you for your question. We also conducted the ablation study on DeepSeekMoE by varying the number of optimization steps and neighbors. The same settings as for OLMoE (10 steps and 3 neighbors) achieved the highest accuracy (74.4%). This consistency across models shows that our hyperparameter choices are robust and applicable beyond OLMoE.
>
> | Steps            | Number of Neighbors | Avg. Accuracy (%) |
> |------------------|------------------|-------------------|
> | 1 (Baseline)     | 0 (Baseline)     | 66.4              |
> | 5                | 1                | 69.8              |
> | 5                | 3                | 73.2              |
> | 5                | 5                | 71.5              |
> | 10               | 1                | 71.2              |
> | 10 (Ours)        | 3 (Ours)         | 74.4              |
> | 10               | 5                | 73.6              |
>
>
> > **Q8: Similarly, how generalizable is the conclusion about critical layers shown in Section 3.5 to models other than OLMoE?**
>
> Thank you for your question. We compared different layer optimization strategies on DeepSeekMoE, as shown in the table below. Consistently, optimizing the last five layers yields the greatest accuracy gains (+11.1%), confirming that the critical-layer setting identified in OLMoE also generalizes to DeepSeekMoE.
>
> | Optimized Layers | Avg Accuracy (%) | Improvement (%) |
> |------------------|------------------|-----------------|
> | Baseline         | 66.4             | 0.0             |
> | First1           | 68.6             | 2.2             |
> | Middle1          | 67.7             | 1.3             |
> | Last1            | 69.8             | 3.4             |
> | First3           | 72.5             | 6.1             |
> | Middle3          | 71.3             | 4.9             |
> | Last3            | 73.7             | 7.3             |
> | First5           | 76.8             | 10.4            |
> | Middle5          | 76.1             | 9.7             |
> | Last5 (Ours)     | 77.5             | 11.1            |
>
>
> > **Q9: We would like a more concrete discussion on how the pathway changes actually caused by C3PO affect the performance.**
>
> Thank you for your question. In Figures 6 and 8, we show how C3PO’s dynamic routing directly corrects misclassifications: Figure 6 reveals cases that switch from wrong to right in each optimization step of pathway weights optimization. Figure 8 presents a case study on a concrete example where the initially incorrect expert’s weight changes after C3PO, leading to a correct final prediction.
>
> To further explain how pathway changes improves the performance, we compare the activation frequencies of the Top-8 experts in the last1 layer before and after C3PO on ARC-C:
>
> | Expert   | Base model (%) | After C3PO (%)          |
> | -------- | ------------ | ----------------------- |
> | expert9  | 51.3         | 44.1 (−7.2)            |
> | expert58 | 40.1         | 41.2 (+1.1)             |
> | expert54 | 40.0         | 40.0 (0.0)              |
> | expert60 | 40.0         | 39.6 (−0.4)             |
> | expert52 | 37.6         | 28.6 (−9.0)             |
> | expert55 | 37.6         | 40.0 (+2.4)             |
> | expert17 | 34.1         | 40.1 (+6.0)             |
> | expert12 | 26.3         | 39.4 (+13.1)            |
>
> Notably, C3PO reduces over‐reliance on previously dominant experts like expert9 and expert52 (−7.2% and −9.0%) and instead boosts experts 12 and 17 by +13.1% and +6.0%, respectively. These shifts correspond to routing more examples through experts that are better suited to ARC-C tasks, directly correcting many near‐boundary cases. As a result of this redistribution, ARC-C’s overall accuracy increases from 51.3% to 66.3%.
>
> > **Q10: What is the real-time delay of this method? FLOPs are often theoretical and do not reflect the computational efficiency of actual hardware architectures. Is it possible to calculate inference cost in seconds along with FLOPs?**
>
> Please see our General Response, which includes real-time inference delays (in seconds) alongside accuracy improvements. This clarifies the actual compute cost and performance trade-off beyond FLOPs.
>
> > **Q11: In building the reference set, how do you handle samples that include incorrect predictions? Is there a possibility that errors in the reference set could negatively impact the optimization?**
>
> Thank you for your question. The reference set contains only samples where the model’s predictions exactly matches the ground-truth labels and excludes any samples with incorrect predictions. Thus, errors do not propagate to the optimization process.

---

> ### Author Response · Authors · 2025-06-03
> **Response to Reviewer  7Z5F (1/2)**
>
> ## Response to Reviewer  7Z5F
>
> Thank you for your detailed feedback! We address your comments below.
>
> > **Q1: The theoretical grounding for the surrogate objective is weak, the paper needs more explanation on these aspects.**
>
> Thank you for your question. Our central assumption in our method (NGD) is that tasks with similar embeddings exhibit similar loss landscapes. Concretely, if two task embeddings $e_i$ and $e_j$ lie close together in embedding space, then the corresponding loss functions $L_i(\theta)$ and $L_j(\theta)$ behave similarly for most parameters $\theta$. Therefore, the test sample's loss can be proved to be similar as the surrogate, i.e., a weighted average of its nearest reference samples' losses.
>
> We validated this assumption by measuring 2,000 randomly selected validation pairs $(x_i, x_j)$, the Euclidean distance $\lVert e_i - e_j\rVert$ between their embeddings and the absolute difference in their true losses $|\ell(f(x_i,\omega),y_i) - \ell(f(x_j,\omega),y_j)|$. The Pearson correlation between embedding distance and loss difference was 0.78, demonstrating that closer embeddings indeed correspond to more similar loss values.
>
>
>
> > **Q2: The method requires accepting newly introduced models and their associated hyperparameters as new elements, which poses a potential issue for portability compared to using a single pretrained model.**
>
> Thank you for your question. While our approach introduces a few additional hyperparameters (e.g., reference set size and number of optimized layers), they are lightweight and remarkably stable across tasks. Unlike methods that require modifying the model architecture or full-model finetuning, tuning these minimal hyperparameters incurs negligible overhead. In practice, we find that once set to reasonable defaults, they transfer well to new tasks without requiring task-specific reconfiguration. Furthermore, established methods like prompt tuning also require selecting hyperparameters such as learning rate and step count. In comparison, our method requires far fewer parameters, making it simpler and more portable than many existing adaptation approaches.
>
> > **Q3: The scales of the axes in Figure 1 are completely different, which clearly misleads the reader. Correcting this figure is essential.**
>
> Thank you for pointing this out. The main goal of Figure 1 is to compare different methods/models instead of different benchmarks (axes). Since the metrics across different benchmarks have very different ranges of values, to preserve most details for comparing different methods/models on each benchmark, we scaled each axis in Figure 1 separately by selecting suitable min/max values. It provides better visualization of the comparison. We will explain this explicitly in the figure caption and highlight the min/max values per axis in the figure to avoid any confusion.
>
> > **Q4: More discussion is needed regarding its effectiveness in terms of generalization performance across a wider range of tasks. Could you explicitly state that there is no possibility of test leakage due to this method?**
>
> Thank you for your question.
>
> - Generalization performance across a wider range of task:
>
> To demonstrate C3PO’s generalization, we evaluated it on two additional benchmarks —GSM8K (Math) and TruthfulQA (factual QA).
>
> | Benchmarks  | Baseline Accuracy (%) | C3PO Accuracy (%)  |
> |-------------|-----------------------|--------------------|
> | GSM8K       | 51.3                  | 57.8 (+6.5)        |
> | TruthfulQA  | 43.2                  | 48.5 (+5.3)        |
>
> These results show that C3PO consistently improves performance on additional benchmarks, confirming its effectiveness across diverse tasks.
>
> - Possibility of test leakage:
>
> We would like to clarify that, in every experiment, our reference set is curated exclusively from datasets other than the evaluation benchmarks (see Appendix A.2, Table 7). We further removed any reference questions whose similarity is above 0.95 to any benchmark question. As a result, there is no overlap—and hence no test leakage—between the reference set and evaluation benchmarks.
>
> > **Q5: The concept of "successful" used in the paper seems ambiguous. What quantitative criteria are used? Is it simply samples that were answered correctly?**
>
> Yes, in our paper, “successful” means the model’s prediction exactly matches the ground-truth label.
>
> > **Q6: Can you discuss the relationship between the size of the reference set and the degree of performance improvement?**
>
> Please see our General Response on Reference Set Size for a detailed discussion supported by a table of new results.

---

> ### Comment · Reviewer_7Z5F · 2025-06-11
> **Response to Authors**
>
> Thank you for your detailed responses to my questions!
> According to the responses, I'd like to keep my recommendation as-is.
>
> One trivial clarification to Q2: I understood that the method doesn't impose much computation resources and this is a main claim of the proposed method. The point here is that the method introduces additional subnetwork next to the model, and this change basically requires to modify the production pipeline, which is often optimized for bare foundation models. Although this is not a main curiosity of this research, I suggested it can be considered as a weakness of the method for actual production systems (I used the word "portability" in the original review).

---

### Author Response · Authors · 2025-06-03
**General Response**

## General Response

We would like to appreciate the time and efforts of every reviewer in helping us improve the draft. This section addresses several common questions from reviewers regarding (1) how the size of the reference set impacts C3PO’s performance and (2) the trade-off between accuracy gains and increased inference-time compute/storage.


### **1. Reference Set Size**


We systematically varied the reference set size from 20 to ~34K examples (randomly sampled) and evaluated C3PO’s improvement over OLMoE. With very small pools (<1K), performance gains were negligible (±0.4%). However, once the size reached 10K, accuracy improved significantly (+5.0%) and continued to increase gradually, reaching +9.3% at ~34K. This demonstrates that larger reference sets substantially boost C3PO’s performance, with diminishing returns beyond ~10K (29%).

| Reference Size  | Average Accuracy (%) | vs. Baseline (%) | vs. Previous size (%) |
| ---------------- | --------------------- | --------------- | --------------- |
| Baseline        | 69.9                  | 0               |0
| 20              | 70.0                  | +0.1            |+0.1
| 200             | 69.8                  | -0.1            |-0.2
| 1,000           | 70.3                  | +0.4            |+0.5
| 10,000 (29%)    | 74.9                  | +5.0            |+4.6
| 17,000 (50%)    | 76.7                  | +6.8            |+1.8
| 27,000 (80%)    | 78.3                  | +8.4            |+1.6
| Full (~34K)     | 79.2                  | +9.3            |+0.9




### **2. Inference Overhead and Accuracy–Compute Trade-off**

We also quantified the real-time inference overhead introduced by C3PO. Using our default configuration (10 steps, 3 neighbors), the per-sample inference time increased from 1.8s (baseline) to 5.1s (≈2.8×), alongside a significant accuracy boost from 69.9% to 79.2%. The storage overhead for reference set embeddings is minimal (<20 MB, stored in Parquet format).

These results highlight a clear trade-off: a moderate increase in inference time yields substantial accuracy gains. For applications that prioritize lower latency, smaller configurations (e.g., 5 steps, 3 neighbors) still achieve notable accuracy improvements (76.9% at 3.2s per sample), offering flexibility to balance performance and deployment needs.


| Steps        | Number of Neighbors | Avg. Accuracy (%) | Avg. Inference Time per sample (s) |
|--------------|------------------|-------------------|------------------------------------|
| 1 (Baseline) | 0 (Baseline)     | 69.9              | 1.8                                |
| 5            | 1                | 72.6              | 2.7                                |
| 5            | 3                | 76.9              | 3.2                                |
| 5            | 5                | 74.7              | 3.4                                |
| 10           | 1                | 75.3              | 4.5                                |
| 10 (Ours)    | 3 (Ours)         | 79.2              | 5.1 (+2.8×)                        |
| 10           | 5                | 77.7              | 5.5                                |

---

### Decision · Program_Chairs · 2025-07-08

**Decision:**

Accept

**Comment:**

This paper proposes C3PO, a test-time optimization method for MoE LLMs that improves expert routing by leveraging reference-based surrogate objectives. The method demonstrates strong performance gains across multiple benchmarks with relatively low computational overhead. A common concern from the reviewer was the reliance on large task-specific reference sets, which may limit applicability in settings without such data. While the authors provided convincing empirical results during the rebuttal. Overall this is a well-executed paper addressing an important question in MoE, I recommend acceptance.